# Perspective on the Role of Gut Microbiome in the Treatment of Hepatocellular Carcinoma with Immune Checkpoint Inhibitors

**DOI:** 10.3390/medicina59081427

**Published:** 2023-08-06

**Authors:** Ludovico Abenavoli, Michele Montori, Gianluca Svegliati Baroni, Maria Eva Argenziano, Francesca Giorgi, Giuseppe Guido Maria Scarlata, Francesca Ponziani, Emidio Scarpellini

**Affiliations:** 1Department of Health Sciences, University Magna Graecia, 88100 Catanzaro, Italy; giuseppeguidomaria.scarlata@unicz.it; 2Clinic of Gastroenterology and Hepatology, Emergency Digestive Endoscopy, Polytechnics University of Marche, 60126 Ancona, Italy; michelemontori93@gmail.com (M.M.); mariaeva.argenziano@gmail.com (M.E.A.); 3“Transplant and Hepatic Damage” Unit, Polytechnics University of Marche, 60126 Ancona, Italy; g.svegliati@univpm.it; 4Oncology Unit, “Madonna del Soccorso” General Hospital, 63074 San Benedetto del Tronto, Italy; francesca.giorgi@sanita.marche.it; 5Digestive Disease Center (C.E.M.A.D.), Fondazione Policlinico Universitario Agostino Gemelli IRCCS, 00168 Rome, Italy; francesca.ponziani@gmail.com; 6Translational Medicine and Surgery Department, Catholic University of the Sacred Heart, 00168 Rome, Italy; 7Translational Research in GastroIntestinal Disorders (T.A.R.G.I.D.), KU Leuven, Herestraat 49, 3000 Leuven, Belgium; emidio.scarpellini@med.kuleuven.be; 8Hepatology Outpatient Clinic, “Madonna del Soccorso” General Hospital, 63074 San Benedetto del Tronto, Italy

**Keywords:** immunotherapy, checkpoint inhibitors, gut microbiota, fecal microbiota transplantation

## Abstract

*Background and Objectives:* Hepatocellular carcinoma (HCC) is the leading cause of liver cancer worldwide and has a high mortality rate. Its incidence has increased due to metabolic-associated liver disease (MAFLD) epidemics. Liver transplantation and surgery remain the most resolute measures. Despite the optimistic use of multi-kinase inhibitors, namely sorafenib, the co-existence of chronic liver disease made the response rate low in these patients. Immune checkpoint inhibitors (ICIs) have become a promising hope for certain advanced solid tumors and, also, for advanced HCC. Unfortunately, a large cohort of patients with HCC fail to respond to immunotherapy. *Materials and Methods:* We conducted a narrative search on the main medical databases for original articles, reviews, meta-analyses, randomized clinical trials, and case series using the following keywords and acronyms and their associations: hepatocellular carcinoma, immunotherapy, checkpoint inhibitors, gut microbiota, and fecal microbiota transplantation. *Results:* ICIs are a promising and sufficiently safe treatment option for HCC. In detail, they have significantly improved survival and prognosis in these patients vs. sorafenib. Although there are several highlighted mechanisms of resistance, the gut microbiota signature can be used both as a response biomarker and as an effect enhancer. Practically, probiotic dose-finding and fecal microbiota transplantation are the weapons that can be used to increase ICI’s treatment-response-reducing resistance mechanisms. *Conclusion:* Immunotherapy has been a significant step-up in HCC treatment, and gut microbiota modulation is an effective liaison to increase its efficacy.

## 1. Introduction

Hepatocellular carcinoma (HCC) is the primary tumor of the liver, often developing in the context of chronic liver disease. It has a global prevalence among cancers, making it the third and seventh most common malignancy in men and women, respectively. Interestingly, HCC is the sixth most commonly occurring cancer worldwide and, due to its constantly increasing incidence, has become the third leading cause of cancer-related death among general populations and the most common cause of death in patients with cirrhosis [1,2]. The incidence of HCC is rapidly growing despite the decreased incidence of chronic hepatitis B virus (HBV) and hepatitis C virus (HCV) falling, mainly because of the new metabolic pandemic affecting our Westernized societies, re-assembled by metabolically associated fatty liver disease (MAFLD) that comprised the definition of nonalcoholic fatty liver disease (NAFLD) [3,4].

Thus, both the increased prevalence and mortality of HCC have been pushing researchers towards better therapeutic approaches. In fact, HCC treatment is complex because of the scarce knowledge of genome mutations and variegated pathophysiology. The standard of care for HCC remains orthotopic liver transplantation and/or surgical resection at the early neoplasm stage [5]. However, we must recognize that there is a shortage of available organs for transplantation and that a high percentage of HCC patients are not eligible for surgical resection as they are not at early cancer stages and, more importantly, they suffer from chronic liver disease leading to advanced organ dysfunction [6,7]. In detail, the most widely adopted HCC staging classification is the updated Barcelona Clinic Liver Cancer (BCLC) [8]. The definitive curative therapies for HCC remain surgical resection and liver transplantation, which can be performed only in patients at very early (0) and early (A) stages. However, because of similar survival times, less invasiveness and, last but not least, a lower economic burden compared to surgical options, percutaneous ablative therapies (namely, radiofrequency ablation (RFA) and microwave ablation (MWA)) are considered the first treatment approach in both of these stages [9,10]. In fact, these techniques have an effective local antitumor effect, but the response rate is relatively weak and might not lead to tumor growth control. Indeed, there is a high local recurrence rate of HCC [11].

It is important to mention that the vast majority of HCC patients (namely, about 65–70%) are still diagnosed in the intermediate (B) or advanced (C) tumoral stages. Therefore, they are ineligible for radical therapies. These patients are considered for transarterial or systemic therapies. The latter show effective results but, at the current status, are non-curative or “palliative”. Precisely, they yield a lower 5-year survival rate. In particular, according to BCLC tumor staging and management, transarterial chemoembolization (TACE) is recommended as first-line therapy for unresectable intermediate-stage HCC (stage B) [12]. More recently, other radiological locoregional therapies have been considered, and other transarterial techniques (namely, transarterial radioembolization (TARE) with yttrium-90) have been suggested as a safe and effective alternative treatment options for HCC patients with a liver-prevalently located disease but not able to tolerate systemic therapies [13].

Despite its poor side effect profile and scarce improvement in overall survival (OS) (namely, less than 3 months vs. placebo), the multi-kinase inhibitor sorafenib has been used as the first-line therapy for Child-Pugh A liver cirrhosis and unresectable/metastatic HCC [7]. Two trials, Sorafenib HCC Assessment Randomized Protocol (SHARP) and Asia Pacific (AP), led to its Food and Drug administration (FDA) approval in 2007 [14,15]. The OS rate of sorafenib is much higher in patients with chronic HCV hepatitis than in those with other etiologies [16]. Subsequently, Lenvatinib was approved as an alternative to sorafenib because it was non-inferior to it [17]. Multi-target tyrosine inhibitors (regorafenib and cabozantinib) [18] and vascular endothelial growth factor (VEGF) receptor inhibitors (ramucirumab) are single-agent second-line treatments for patients failing to respond to sorafenib [19,20]. The combination of atezolizumab and bevacizumab is now regarded as the standard first-line treatment for patients with advanced HCC due to the significant and clinically meaningful improvements in terms of OS, progression-free survival (PFS), objective response rate (ORR) [8,21], and complete response rate (CRR) compared with sorafenib monotherapy [22]. Indeed, the combination of tremelimumab and durvalumab has been reported to be superior to sorafenib in patients with advanced or unresectable HCC, adding another first-line treatment option [23].

Recently, immune checkpoint inhibitors (ICIs) have emerged as alternatives for patients with adequate performance status. In fact, HCC cells have a deep immune system surveillance and escape behavior [24]. In 2017, the FDA approved nivolumab as an add-on treatment for patients failing to respond to sorafenib. Therefore, pembrolizumab was approved. These two immunotherapies belong to the group of programmed cell death protein-1 (PD-1) inhibitors. In addition, a combination of ipilimumab [a cytotoxic T-lymphocyte-associated protein 4 (CTLA-4) inhibitor] and nivolumab was also approved by the FDA. These trials showed one-by-one that ICIs are superior to sorafenib in terms of OS and PFS [25]. Despite these significant improvements in immunotherapy vs. the standard of care for HCC treatment, almost 60% of these patients do not respond to ICIs. In detail, because of several restricted selection parameters, only 10–20% of HCC patients are eligible for first-line ICI therapy. Moreover, this eligible percentage is reduced to less than 10% in the second-line treatment. Therefore, only a small number of HCC patients could actually benefit from immunotherapy. Thus, there is an urgent need for effective predictive serological and/or tissue biomarkers to identify patients likely to benefit from immunotherapy in the context of “personalized “therapy choice. This would reduce the economic impact of treatments’ costs on our healthcare systems. The use of effective biomarkers would also help to avoid ICI-associated adverse events in patients pre-identified as non-responders [26,27].

Among the emerging biomarkers of both treatment response and adverse event prediction, the human gut microbiota is gaining more and more favorable evidence [28,29]. The human gut microbiota is a complex ecosystem encompassing more than 50 bacterial species shared by every individual. Indeed, it also includes viruses, protozoa, fungi, archaea, and yeasts [30]. Its functions range from nutrient absorption and digestion to metabolism modulation and immune system regulation [31]. The latter is of particular interest in the context of carcinogenesis and the re-establishment of immune surveillance in cancerous conditions [32].

We aimed to review literature data on HCC treatment options and, in particular, the impact of ICIs, their use limitations, and the gut microbiota’s role as a response biomarker and, perhaps, an enhancer.

## 2. Methods: Literature Search

In particular, we conducted a PubMed and Medline search for original articles, reviews, meta-analyses, and case series using the following keywords, their acronyms, and their associations: hepatocellular carcinoma, immunotherapy, checkpoint inhibitors, gut microbiota, and fecal microbiota transplantation. When appropriate, preliminary pieces of evidence from abstracts belonging to main national and international gastroenterological meetings (e.g., United European Gastroenterology Week, Digestive Disease Week) were also included. The items found from the above-mentioned sources were reviewed by two of the authors (L.A. and M.M.) according to PRISMA guidelines [33]. The last MEDLINE search was dated 28 February 2023. Finally, a narrative review was performed.

### 2.1. Hepatocellular Carcinoma Treatment

#### 2.1.1. HCC Treatment Fundaments

In brief, clinical guidelines for the standard of care of HCC patients include: curative therapies (e.g., radiofrequency or microwave ablation, liver resection, and transplantation) for early-stage cancers; transarterial chemoembolization (TACE) for intermediate-stage cancers; and, the main item treated in this review of literature, systemic pharmacologic treatments reserved for advanced tumors. The latter can be divided into the first and second lines [34,35]. Interestingly, the median survival time is higher than 6 years for resection/ablation procedures and 10 years for transplantation. At intermediate HCC progression stages, there is a median survival time of 20–30 months. The latter is limited to 10–16 months for advanced HCC staging [22,36]. Thus, the advanced tumor stages are fertile ground for research trying to develop systemic therapy approaches for these patients.

The SHARP trial and consequent approval of sorafenib use in advanced HCCs have been cutting-edge moments in hepatology. The main success was the recognition of overall survival (OS) benefits with sorafenib use. ICIs have overcome these limits. Certainly, systemic therapy is for those patients not eligible for locoregional/curative therapy but with adequate performance status and stable liver function. Preliminarily, patients should be screened for viral hepatitis and predictive markers of response, as stated by the American Society of Clinical Oncology (ASCO) [21,37]. Endpoints of HCC therapies are overall survival and other surrogate endpoints (e.g., response rate and progression-free survival (PFS)) [21,38], following the RECIST (the Response Evaluation Criteria in Solid Tumors) and, thereafter, the modified RECIST (mRECIST) guidelines [21,39]. Importantly, sorafenib obtained a significant survival extension but at a low objective response rate (ORR), according to standard RECIST criteria.

#### 2.1.2. Immune Checkpoints and Hepatocellular Carcinoma: The Origin of the New Frontier of Immunotherapy

Immune checkpoints are expressed in various cell types, such as natural killer (NK) and dendritic cells (DCs), tumor-associated macrophages (TAMs), monocytes, and myeloid-derived suppressor cells (MDSCs) [40]. Immune checkpoints are proteins that can inhibit immune cell function, leading to a reduction in wide-field tissue damage. Despite that, as already demonstrated in the literature, tumor cells may disrupt the immune resistance mechanism [41]. In human cancers, the most studied immune checkpoints are: cytotoxic T-lymphocyte associated protein 4 (CTLA-4), programmed death cell protein 1 (PD1)/ligands (PDL1), which showed to have an interesting role in HCC treatment [42]; lymphocyte activation gene 3 (LAG-3), T-cell membrane protein 3 (TIM-3), and B- and T-lymphocyte attenuator (BTLA). The last three molecules do not fall within the field of interest of this review [43].

HCC appeared to be a good environment for immune checkpoint inhibitors (ICIs) treatment due to the high intrahepatic lymphocyte expression of PD-1 in chronic liver diseases. In detail, these diseases are associated with a greater expression of PD-1L in Kupffer cells [44]. In further detail, PD-1 is a receptor expressed on activated T cells, B cells, NK cells, MDSC, and DCs. Mainly, it can inhibit the immune system via tyrosine phosphatase SHP-2, preventing autoimmunity [45]. The ligand is PD-L1, which is expressed by somatic cells in a pro-inflammatory setup; it also suppresses T-cell migration, proliferation, and the release of cytotoxic cytokines (Figure 1A) [46]. Therefore, it is of extreme importance for signaling in liver tumors: It is driven by cancer cells, which constantly express PD-L1 and consequently activate PD-1 in tumor-infiltrating lymphocytes (TILs), evading immune surveillance [47]. Interesting, several studies demonstrated the association between high expression of PD-L1 in liver cancer cells and poor prognosis in HCC [48,49]. The latter was mainly due to tumor recurrence aggressiveness [50,51].

CTLA4 is a negative regulator of the immune response. Molecularly, it is an intracellular protein within T-cells and translocates to the cell surface when the T-cell receptor is binding CD28. Furthermore, surface CTLA4 binds CD80 and CD86, blocking the linkage of these to CD28. This results in inhibition of T cell proliferation and activation [52]. Importantly, its role in tumorigenesis is associated with the inhibition of interaction between T cells and antigen-presenting cells. Thus, there is a reduction in cytokine production (e.g., IL-2) and T cell proliferation [53]. On the other hand, CTLA-4 may stimulate the expression of immune regulatory cytokines such as transforming growth factor-β (TGF-β). Therefore, CTLA4 has a role in T-reg activation and differentiation because its receptor is constitutively expressed on these cells (Figure 1B). Further, when it is blocked, antitumor activity and autoimmunity are impaired [54]. Therapeutically, the anti-CTLA-4 antibodies can block CTLA4 on Tregs together while enhancing T(eff) cell functioning [55].

#### 2.1.3. Immunotherapy and HCC

Since 2008, the main treatment for hepatocellular carcinoma (HCC) has been the oral multi-kinase inhibitor sorafenib. It has proven to prolong the overall survival time by 2.8 months [14]. During the last few years, they have used several combinations of sorafenib and similar molecules, such as lenvatinib (an inhibitor of VEGF receptors 1–3, FGF receptors 1–4, and the PDGF receptor α). These studies showed non-inferiority in overall survival vs. untreated advanced hepatocellular carcinoma patients [17]. Recently, a lot of interest has focused on immune-checkpoint inhibitors targeting CTLA4 [56,57] and PD1/PDL1 [58], respectively.

More in detail, two different molecules (namely, PD1 inhibitors), already approved for patients with advanced or metastatic melanoma and metastatic refractory non-small cell lung cancer [59,60], have shown a promising efficacy profile for HCC treatment. Nivolumab (anti-PD-1) was approved in 2016 and demonstrated to reach an objective response rate (ORR) of about 20% [61]. Importantly, the latter is fourfold bigger than those of sorafenib. Moreover, in a phase III multicenter trial, nivolumab was compared with sorafenib as a first-line treatment in patients with advanced HCC. Although the OS was not statistically significant (median OS of 16.4 vs. 14.7 months for nivolumab and sorafenib, respectively), nivolumab was shown to be safe and to have clinical activity improvement. Therefore, this drug may be considered as a first-line treatment for patients in whom TKI and anti-angiogenetic drugs are contraindicated [62]. Pembrolizumab (anti-PD-1) was approved in 2018 after a phase 2 multicenter trial demonstrated it to be able to increase the oncologic response in patients with advanced hepatocellular carcinoma previously treated with sorafenib (ORR 17%, 44% with stable disease, 33% with progressive disease) [63]. Similar results were obtained in a phase III trial comparing pembrolizumab vs. placebo as second-line therapy for sorafenib-pretreated patients. However, in this investigation, OS was not statistically significant (13.9 months vs. 10.6 months, pembrolizumab vs. placebo group, respectively). Indeed, the ORR was stable at 17% [64].

To date, there are several ongoing clinical trials (namely, phase I/II) investigating other PD-L1 inhibitors (namely, avelumab, atezolizumab, and durvalumab) used as monotherapy or in combination with other ICIs [15,16]. In particular, a multicenter non-randomized phase 2 trial showed promising results when these PD-L1 inhibitors were used as first- and second-line treatment of advanced HCC with the combination of camrelizumab (anti-PD1) and apatinib (VEGFR-2 tyrosine kinase inhibitor). In this case, the ORR was 34.3% when PD-L1 inhibitors were administered as first-line therapy and 22.5% when administered as second-line therapy. Interestingly, the PFS was 5.7 months and 5.5 months, respectively [30,31]. Indeed, the most important breakthrough in systemic therapy for advanced HCC has been represented by the combination of atezolizumab (anti-PDL1)/bevacizumab (anti-VEGFA). This is the updated first-line treatment for hepatocellular carcinoma due to the proven greater OSS (19.2 months) [14] and PFS (6.8 months), respectively [15,16] (Table 1).

Unfortunately, despite these very promising findings, a significant portion of HCC patients show no significant benefits from immune-therapy administration. These contrasting results can be explained by different tumor biology, characteristics, and etiologies [22,65]. For instance, it is interesting to report findings from an exploratory analysis of the IMbrave150 trial. The latter showed the combination therapy of atezolizumab and bevacizumab having an ORR of 27% in NASH-HCC vs. 35% in HCC of other etiologies [66]. More interestingly, immune therapy shows a significantly different outcome for patients with viral vs. non-viral HCC origins. In fact, a recent meta-analysis confirmed that ICIs are less effective in patients with non-viral-derived HCC [67]. Indeed, this evidence is not completely in agreement. Two further meta-analyses reported no difference in ORR for viral vs. non-viral HCC [68,69].

The second class of molecules currently under investigation for HCC treatment is the CTLA-4 inhibitors (namely, tremelimumab and ipilimumab). Tremelimumab was initially studied in a phase II clinical trial as monotherapy for patients with HCV-related cirrhosis and secondary HCC, demonstrating an ORR of 17.6%. Time to progression was 6.48 months with a good safety profile [70]. Subsequently, another clinical trial studied the combination of tremelimumab with locoregional treatment (namely, TACE or radiofrequency ablation (RIA)), showing a higher ORR of 26.3%. This could be explained by the accumulation of intra-tumoral CD8+ T cells in treated patients. Collaterally, a reduction in HCV viral load was observed [71]. More recently, the phase III HIMALAYA trial demonstrated the greater efficacy of the combination of durvalumab (anti-PD-L1) and tremelimumab (STRIDE) vs. sorafenib as first line therapy for patients with advanced HCC. In fact, STRIDE showed a 36-month OS rate of 30.7% vs. 20.2% for sorafenib. This finding was accompanied by a manageable safety profile [72]. Thus, STRIDE has been recently approved by the US FDA for HCC treatment. This step paves the way for a new scenario for first-line treatment of advanced HCC [73,74].

Currently, there is another randomized, multicenter phase III trial using the combination of nivolumab and ipilimumab vs. sorafenib or lenvatinib as first-line treatment for HCC. This trial follows a previous study showing this combination therapy to have an acceptable safety profile and an interesting ORR of 31% vs. 14% for nivolumab monotherapy [75]. However, ICI use in HCC shows a significant rate of non-responders. There are several mechanisms explaining this finding. They are shown in Table 2.

#### 2.1.4. Fecal Microbiota Transplantation and Immunotherapy in HCC: Beyond a Simple Cancer-Intestinal Bacteria Association

The rising knowledge on gut microbiota functions, in general, and the contribution of gut “dysbiosis” in the loss of barrier function, in particular, leading to altered “gut–liver axis” and gut–liver immune system dysfunction, are the basis for future treatment aimed to block fibrosis progression in liver cirrhosis and to treat and destroy HCC cells [88,89].

Specifically, the mechanistic knowledge of the microbiome-HCC harmful “game” derives from preclinical animal models. In rodents, the activation of TLR4 signaling after gram-negative lipopolysaccharide (LPS) exposure and, also, the direct detrimental effect of microbially produced secondary bile acids within the liver “promote” carcinogenesis [90]. Therefore, Dapito et al. showed that the depletion of the microbiota “protected “against fibrosis and cancer development in mice [91]. Similarly, neomycin protected against HCC development upon diethylnitrosamine/carbon tetrachloride (DEN/CCl4) animal administration [92]. Secondary bile acids, namely deoxycholic acid (DCA), contribute to liver inflammation via the promotion of the senescence-associated secretory phenotype (SASP). This pathway is crucial in metabolic (namely, obesity)-associated HCC development [80]. In addition, DCA is able to induce NASH-associated HCC via mTOR activation [93]. Conversely, antibiotic treatment led to a reduction of secondary bile acid and, consensually, an increase in the primary bile acid pool, resulting in increased anti-tumor immunity expansion [94].

Thus, the dreamt bacteriotherapy can restore “gut eubiosis,” re-establishing physiological intestinal permeability and reducing the passage of pathogen-associated molecular patterns (PAMPs) such as endotoxin and the related chronic inflammation within the liver. The latter effect can re-establish immune surveillance towards hepatocytes accumulating mutations, namely potential tumor cells (e.g., HCC and its “brothers “) [75]. We can hypothesize that bacteriotherapy would be as effective as it was earlier, namely during the first stages of chronic liver disease [95].

Gut microbiota modulation is feasible through diet, probiotics, prebiotics, and antibiotics [96]. The latter have obtained some promising results in terms of efficacy in preventing HCC development, but these are not devoid of toxic and resistance-development side effects [83,84]. Probiotic administration in HCC patients has shown safer and efficacy-proven results from the few available trials. In a randomized clinical trial, probiotics’ capability to re-establish gut barrier function in F3-F4 HCC subjects undergoing surgery was proven [97]. Furthermore, using the probiotic BIFICO throughout the preoperative phase of HCC patients’, the authors found these bugs able to accelerate postoperative liver function recovery [98]. Indeed, the use of probiotics in liver cancer patients needs more evidence and must take into consideration the issue of dose- and time-finding. Moreover, we cannot exclude the development of potentially pathogenic strains. Thus, a new direct and more effective way to restore gut eubiosis and potentially increase ICI efficacy is fecal microbiota transplantation (FMT).

In FMT, fecal healthy donors are carefully selected through strict exclusion criteria (e.g., malnutrition, obesity) in order to exclude the risk of disease transmission such as in the obesity case [99]. Technically, fecal material is collected from the donor, suspended in a saline solution, and mixed in a blender. It results in liquefied stool that is filtered through a strainer in order to remove fibers [100]. Therefore, fecal material is ready to be delivered via endoscopy (e.g., colonoscopy or nasojejunal tube), enema, or colonic transendoscopic enteral tubing. Oral capsules have shown similar efficacy as colonoscopy-administered procedures. However, the frequency of doses and optimal overall duration of the capsule-administration regimen are still under investigation [101]. To date, FMT has been approved for the treatment of recurrent *Clostridium difficile* infection by 2014, with an effectiveness of about 90% [102].

There are promising studies on its use in cancer conditions, but not in HCC patients yet. Baruch et al. reported the first-in-human clinical trials in melanoma patients. FMT was significantly associated with an immune system switch towards immune surveillance, as described by changes in gene expression profiles in both the gut lamina propria and the tumor neighborhood [103].

#### 2.1.5. Gut Microbiome and Immunotherapy: A “Navigator” for HCC Treatment

The biopsy-sparing diagnostic approach to HCC has led to a non-invasive prognostic biomarker need. Furthermore, the ICIs arrival has recalled these biomarkers. In addition, despite the promising results obtained with ICIs’ use, a significant rate of patients do not take benefit from immunotherapy. Thus, there is a strong need for “precise “predictive markers in HCC patients [54,65]. Gut microbiota profiling seems a promising HCC treatment-response non-invasive biomarker [104,105].

Zheng et al. have studied dynamic variation and features of fecal gut microbiota during anti-PD1 immunotherapy (namely, Camrelizumab) in HCC after progression on sorafenib. They evaluated differences among responders and non-responders’ fecal samples at baseline, 1 week after treatment, and every 3 weeks until disease progression. Interestingly, fecal samples of responders have higher taxa richness and higher gene counts than those of non-responders. More intriguingly, the inter-group dissimilarity became significantly higher than the intra-group differentiation as early as 6 weeks after treatment imitation.

At baseline, in both responders and non-responders, *Bacteroidetes* were the most abundant, followed by *Firmicutes* and *Proteobacteria*. Typically, this microbial composition remained relatively stable at the phylum level in responders. On the other hand, *Proteobacteria* concentrations already increased after 3 weeks of treatment. They became predominant at week 12 in non-responders. *Proteobacteria’s* increased abundance was explained by the prevalence of *Escherichia coli*. Conversely, the most abundant proteobacterial member in responders was *Klebsiella pneumoniae*. Finally, at the linear discriminant analysis-effect size algorithm, 20 responder-enriched and 15 non-responder-enriched species were identified. In the responder group, there were 4 identified *Lactobacillus* species (namely, *L. oris*, *L. mucosae*, *L. gasseri*, and *L. vaginalis*). *Bifidobacterium dentium*, *Streptococcus thermophilus*, *Coprococcus comes*, *Bacteroides cellulosilyticus*, *Subdoligranulum* sp. *Lachnospiraceae bacterium 7 1 58FAA*, *Ruminococcus obeum*, *Ruminococcus bromii,* and *Akkermansia muciniphila* were also observed in treatment responders [106].

Chung et al. examined the gut microbiome of 8 advanced HCC patients (of whom 6 had chronic hepatitis B). Fecal samples were collected before the first administration of nivolumab and then, at the time of disease progression, during treatment. In responders, fecal samples were collected after 5–7 months of treatment. Responders have a higher Shannon index and a different phylogenetic diversity at the beta diversity analysis, when compared to non-responders. In detail, *Dialister pneumosintes*, *Escherichia coli*, *Lactobacillus reteri*, *Streptococcus mutans*, *Enterococcus faecium*, *Streptococcus gordonii*, *Veillonella atypica*, *Granulicatella* sp., and *Trchuris trichiura* were specifically prevalent in non-responders. *Citrobacter freundii*, *Azospirillum* sp., and *Enterococcus durans* were prevalent in responders. Moreover, an altered *Firmicutes/Bacteroidetes* ratio (<0.5 or >1.5) and a low *Prevotella/Bacteroides* ratio were significantly correlated with the non-responder profile. Conversely, the presence of *Akkermansia* species was observed in responders [107].

Mao et al. studied fecal samples from 65 patients affected by advanced HCC or biliary tract cancer (namely, 30:35) receiving anti-PD-1 therapies. Seventy-four taxa were significantly enriched in responders, compared to 40 taxa in non-responders. Within the first group, there was a higher abundance of *Lachnospiraceae bacterium-GAM79*, *Alistipes* sp. *Marseille-P5997, Ruminococcus calidus*, and *Erysipelotichaceae bacterium-GAM147*. In the non-responder’s group, there was a higher abundance of *Veillonellaceae*. Interestingly, immunotherapy-related adverse events correlated with the phylogenetic diversity of the gut microbiota. This finding can be explained by the immunotherapy-related colitis that was more likely associated with decreased gut microbiome diversity and relative abundance. Sixteen enriched taxa were identified in patients with diarrhea (namely, the *Negativicutes* class, *Veillonellaceae* family, and *Dialister* genus). Particularly, enrichment of *Prevotellamassilia timonensis* was observed in patients with severe diarrhea [108].

Ponziani et al. evaluated prospectively eleven patients with HCC treated with Tremelimumab and/or Durvalumab. Responders showed lower pretreatment fecal calprotectin, an increased relative abundance of *Akkermansia*, and a reduced relative abundance of *Enterobacteriaceae* vs. non-responders. Further, dynamic analysis of fecal calprotectin showed a temporal evolution opposite to the Akkermansia to Enterobacteriaceae ratio and gut microbiota alpha diversity [109].

Lee et al. analyzed baseline fecal samples of 94 patients receiving ICI treatment (nivolumab and pembrolizumab) for HCC (63.4% of those were HBV-related). *Prevotella 9* was enriched in non-responders, whereas *Lachnoclostridium, Lachnospiraceae*, and *Veillonella* were predominant in responders. Furthermore, the evidence of *Lachnoclostridium* enrichment and *Prevotella 9* depletion significantly predicted overall survival. The study included a validation cohort in which a better progression-free survival (PFS) and OS were observed in patients who had a preferable microbial signature, namely depleted *Prevotella 9* and enriched *Lachnoclostridium*, vs. patients with a poor signature (namely, coexistence of enriched Prevotella 9 and depleted Lachnoclostridium) or a fair signature (namely, coexistent depletion or enrichment of these two taxa) [110].

Li et al. also evaluated the oral and gut microbiome profiles of 65 patients with HCC receiving ICIs. They found that *Clostridiales/Ruminococcaceae* were enriched in responders, and *Bacteroidales* were enriched in non-responders. Moreover, patients with a high *Faecalibacterium* abundance had a significantly prolonged PFS vs. those with a low abundance. On the other hand, patients with a high abundance of *Bacteroidales* had a shortened PFS vs. those with a low abundance [111]. The evidence on gut microbiota changes associated with the ICI treatment response are summarized in Table 3.

## 3. Conclusions

HCC is a systemic cancer with growing prevalence and mortality. Despite the encouraging step-up in systemic therapy achieved with sorafenib and improved survival time, there was a big issue concerning tumor progression. The latter has not been solved by further systemic treatment options. More recently, HCC has become a target for novel immune-checkpoint inhibitors. These have shown superiority vs. the traditional multi-kinase inhibitor sorafenib in terms of survival rate and blockage of tumor progression. However, a significant proportion of treated HCC patients do not respond to ICIs. Another limitation of using ICIs is the small number of patients who can be enrolled in immune therapy. Finally, the occurrence of unfavorable side effects is responsible for the interruption of treatment.

Drug-resistance mechanisms, immune system response escape, and unfavorable immune system function within the liver can explain these pitfalls. These are hot topics in future HCC and immune-therapy research.

Beside alpha-fetoprotein, other biomarkers of treatment response have been studied and proposed. Among these, there is the gut microbiota, whose signature has shown interesting findings via new metagenomic methods. More in detail, gut dysbiosis seems to be associated with ICIs’ poor responses. In fact, immune system depression within the liver is associated with gut microbiota derangements. Moreover, certain “eu-” or “dysbiotic” microbiota is associated with a better or worse ICI response, respectively. This evidence is the basis for future lines of research: gut microbiota finger-printing before, during, and after ICI’s treatment can help predict patients’ eligibility, performance, and prognosis; gut dysbiosis modulation can help improve treatment response.

Thus, potential remodulation of dysbiosis via probiotics can improve patients’ outcomes under ICIs. However, this method of microbial modulation has several open issues, including the timing of probiotics’ administration, duration of administration, and side effect profile.

A more direct method to modulate the gut microbiota is FMT. However, data on FMT use in HCC patients are ongoing and call for researchers’ attention. In this regard, we have several concerns: What HCC patient should be treated, and for how long? What should be the safety profile of FMT? What is its interaction with ICIs?

Our attention is therefore focused on the future larger tracing of gut microbiota asset “per patient” in the context of personalized medicine, perhaps using the power of big data analysis provided by artificial intelligence (AI) in medicine.

## Figures and Tables

**Figure 1 medicina-59-01427-f001:**
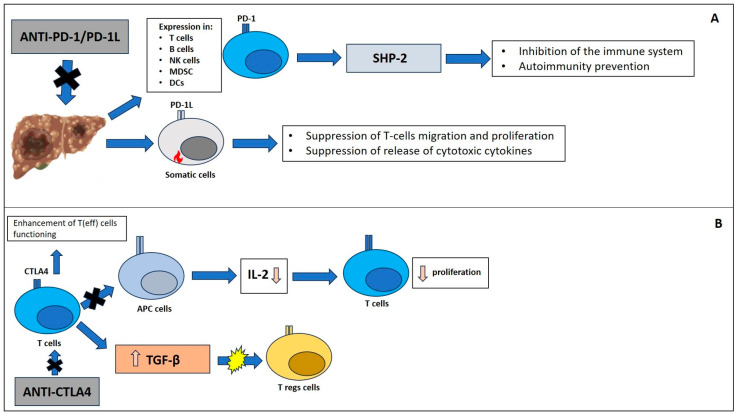
Immune checkpoint pathways: (**A**) PD-1/PD-1L and CTLA4 (**B**).

**Table 1 medicina-59-01427-t001:** Clinical trials on HCC and checkpoint inhibitors.

Trial Name	Experimental Design	Median OS (HR, 95% CI)	Median PFS (HR, 95% CI)	ORR	Reference
IMbrave150	Phase 3 RCT Atezolizumab + Bevacizumab (*n* = 336) vs. Sorafenib (*n* = 165)	19.2 vs. 13.4 months (0.66, 0.52–0.85; *p* < 0.001)	6.9 vs. 4.3 months (0.65, 0.53–0.81; *p* < 0.001)	30% vs. 11% (*p* < 0.001)	[56]
ORIENT-32	Phase III RCT Sintilimab + IBI305 (bevacizumab biosimilar) (*n* = 380) vs. Sorafenib (*n* = 191)	NE vs. 10.4 months (0.57, 0.43–0.75; *p* < 0.0001)	4.6 vs. 2.8 months (0.56, 0.46–0.70; *p* < 0.0001)	21% vs. 4% (*p* < 0.0001)	[21]
CheckMate 459	Phase III RCT Nivolumab (*n* = 371) vs. Sorafenib (*n* = 372)	16.4 vs. 14.7 months (0.85, 0.72–1.02; *p* = 0.075)	3.7 vs. 3.8 months (0.93, 0.79–1.10; *p* = NS)	15% vs. 7% (*p* = NR)	[51]
Cosmic 312	Phase III RCT Atezolizumab + Cabozantinib (*n* = 472) Vs. Sorafenib (*n* = 217)	15.4 vs. 15.5 months (0.90, 0.69–1.18; *p* = 0.44)	6.8 vs. 4.2 months (0.63, 0.44–0.91; *p* = 0.0012)	11% vs. 4% (*p* = NR)	[56]
HIMALAYA	Phase III Tremelimumab + Durvalumab (STRIDE) (*n* = 393); Durvalumab (*n* = 389), both vs. Sorafenib (*n* = 389)	16.4 vs. 13.8 months (0.78, 0.65–0.93; *p* = 0.0035); 16.4 vs. 16.6 months (0.86, 0.73–1.03; noninferiority margin 1.08)	3.8 vs. 4.1 months (0.90, 0.75–1.05; *p* = NR); 3.7 vs. 4.1 months (1.02, 0.88–1.19; *p* = NR)	17.0% vs. 5.1% (*p* = NR); 26.1% vs. 17.5% (*p* = NR)	[14]
LEAP-002	Phase III RCT Lenvatinib + Pembrolizumab (*n* = 395) vs. Levatinib (*n* = 399)	21.2 vs. 19.0 months (0.840, 0.708–0.997; *p* = 0.0227)	8.2 vs. 8.0 months (0.867, 0.734–1.024; *p* = 0.0466)	26.1% vs. 17.5% (*p* = NR)	[21,22]
RATIONALE-301	Phase III RCT Tislelizumab (*n* = 342) vs. Sorafenib (*n* = 332)	15.9 vs. 14.1 months (0.85, 0.71–1.02; *p* = NR)	2.2 vs. 3.5 months (1.10, 0.92–1.33; *p* = NR)	14.3% vs. 5.4% (*p* = NR)	[22]
NCT03764293	Phase III RCT Camrelizumab + Rivoceranib (Apatinib)Vs. Sorafenib (*n* = 271)	22.1 vs. 15.2 months (0.62, 0.49–0.80; *p* < 0.0001)	5.6 vs. 3.7 months (0.52, 0.41–0.65; *p* < 0.0001)	25.4% vs. 5.9% (*p* < 0.0001)	[23,24]
KEYNOTE-224	Phase II non-RCT Pembrolizumab (*n* = 104)	12.9 months	4.9 months	17%	[52]
KEYNOTE-240	Phase III placebo RCT Pembrolizumab (*n* = 278) vs. placebo (*n* = 135)	13.9 vs. 10.6 months (0.781, 0.611–0.998; *p* = 0.0238)	3.0 vs. 2.8 months (0.718, 0.570–0.904; *p* = 0.0022)	18.3% vs. 4.4% (*p* = 0.00007)	[53]
KEYNOTE-394	Phase III placebo RCT Pembrolizumab (*n* = 300) vs. placebo (*n* = 273)	14.6 vs. 13.0 months (0.79, 0.63–0.99; *p* = 0.018)	2.6 vs. 2.3 months (0.74, 0.60–0.92; *p* = 0.0032)	12.7% vs. 1.3% (*p* = 0.00004)	[27]

Note: RCT, randomized clinical trial; NR, not reported; NS, not significant; ORR, objective response rate; OS, overall survival; PFS, progression-free survival; NE, not estimable.

**Table 2 medicina-59-01427-t002:** Mechanisms involved in resistance to ICIs.

HCC-Related Resistance Factors	Extra-HCC Resistance Factors
- Down-regulation of antigen processing: HLA and beta2-microglobulin deletion [76,77]; reduced production of cytokines: loss of *JAK1/2* functioning [78], deletion of *IFNGR1/2*, *IRF1* [79] - *CTNNB1* gene mutation resulting in beta-catenin activation [80]	- PTEN deletion and VEGF upregulation leading to TILs exclusion [81] - Expression of alternative coinhibitory checkpoint receptors (e.g., TIM-3, LAG-3, TIGIT, VISTA and BTLA) [82] - Decreased TILs to Treg ratio [83,84] - Down-regulation of dendritic cell recruitment through b-catenin signaling [85] - Increased immunosuppressive cells expression (e.g., MDSCs, Tregs) [86] - Increased epithelial-to-mesenchymal cells transition [87] - Gut dysbiosis.

Note: HLA: human leukocyte antigen; JAK: janus kinase; IFN: interferon; CTNNB1: catenin beta-1; PTEN: Phosphatase and TENsin; VEGF: Vascular endothelial growth factor; TIL: tumor-infiltrating lymphocytes; TIM-3: T-cell immunoglobulin domain and mucin domain containing molecule-3; LAG-3: lymphocyte-associated gene 3; TIGIT: T cell immunoreceptor with Ig and ITIM domains; VISTA: V-domain Ig suppressor of T-cell activation; BTLA: B- and T-lymphocyte attenuator; MDSC: myeloid derived suppressor cell.

**Table 3 medicina-59-01427-t003:** Gut microbiota changes associated with ICI treatment response profile.

HCC-ICIs Responder(s)	HCC-ICIs Non-Responder(s)
- Diversity level: ↑ [84,89] - Phylum level: ↑ *Firmicutes* [86,91] - Order level: ↑ *Bacteroidales*, ↑ *Clostridiales* [84,89] - Family level: ↑ *Lachnospiraceae*, ↑ *Ruminococcaceae* [89] - Genus level: ↑ *Fecalibacterium*; [84,112] - Species level: ↑ *Akkermansia muciniphila;* ↑ *Bifidobacterium dentium*, ↑ *Blautia obeum*, ↑ *Lachnospiraceae bacterium 7_1_58FAA*, ↑ *Lactobacillus gasseri,* ↑ *Lactobacillus oris*, ↑ *Lactobacillus vaginalis*, ↑ *Lactobacillus. Mucosae*, ↑ *Ruminococcus bromii*, ↑ *Streptococcus thermophiles* [84]	- Diversity level: ↓ [84,89] - Phylum level: ↑ *Proteobacteria* [84,89] - Order level: ↑ *Bacteroidales* [84,89] - Family level: not yet available data - Genus level: not yet available data - Species level: ↑ *Bacteriodes eggerthii*, ↑ *Bacteroides nordii*, ↑ *Escherichia coli*, ↑ *Fusobacterium varium* [84]

## Data Availability

No new data were created or analyzed in this study. Data sharing is not applicable to this article.

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
