# Peer review of "Perspective on the Role of Gut Microbiome in the Treatment of Hepatocellular Carcinoma with Immune Checkpoint Inhibitors"

_medicina, 2023, doi:10.3390/medicina59081427_

Round 1

Reviewer 1 Report

Overall, the article covers an important and timely topic in the field of oncology and provides valuable insights into the potential role of ICIs and gut microbiota in the treatment of HCC.

The article could expand on the limitations and challenges associated with using ICIs and manipulating the gut microbiota in HCC treatment. This would provide a more comprehensive understanding of the topic and guide future research directions.

I suggest to the authors consider the citation of this review article for anti-angiogenesis receptor tyrosine kinase inhibitors. https://link.springer.com/article/10.1007/s11030-022-10406-8

Minor editing of the English language is required.

Author Response

Referee 1

Overall, the article covers an important and timely topic in the field of oncology and provides valuable insights into the potential role of ICIs and gut microbiota in the treatment of HCC.

The article could expand on the limitations and challenges associated with using ICIs and manipulating the gut microbiota in HCC treatment. This would provide a more comprehensive understanding of the topic and guide future research directions.

We thank the reviewer for this precise suggestion: we have highlighted the limitations related to the use of ICIs; we have added some evidence on the limitations and challenges linked to the gut microbiota in HCC treatment in the Conclusions section.

I suggest to the authors consider the citation of this review article for anti-angiogenesis receptor tyrosine kinase inhibitors. https://link.springer.com/article/10.1007/s11030-022-10406-8

We have added the suggested reference.

Minor editing of the English language is required.

We have operated fair English language editing.

Reviewer 2 Report

Dear Authors

I would like to thank you for the opportunity to review this interesting paper focused on a very remarkable and challenging topic that is a lively argument also in daily clinical practice. 

Immunotherapy has remarkably revolutionized the management of advanced HCC and prompted clinical trials, with therapeutic agents being used to selectively target immune cells rather than cancer cells. Despite the encouraging results, many unanswered questions still remain, including which immunotherapy and locoregional treatment can guarantee the best survival and clinical outcomes, the most effective timing and sequence to obtain the most effective therapeutic response and which biological and/or genetic biomarkers can be used to identify patients likely to benefit from this combined approach. In particular, considering the potential modulatory role of human microbiota on antitumoral immune response, the additional administration of specific probiotics as well as the more complex fecal microbiota transplantation in non-responder patients are currently being investi-gated, with rather promising preliminary results.

This paper is pleasurable to read, although it suffers from some limitations that Authors can easily adjust to slightly improve their review making it more eligible for this important Journal. Furthermore, the Authors can improve some sections of the paper, adding information and including other important references about this topic that, in my opinion, should be cited and discussed. 

First of all, despite the title being correct, I suggest improving it. For example: “Perspective on the role of gut microbiome in the treatment of Hepatocellular carcinoma with Immune Checkpoint Inhibitors” or something similar.

In the Introduction, line 46 “HCC is the fourth cancer’s cause of death worldwide”. This is not true. According to the most updated data, HCC is the sixth most commonly occurring cancer worldwide and, due to its constantly increasing incidence, has become the third leading cause of cancer-related death among general populations. Moreover, it represents the most common cause of death in patients with cirrhosis. [https://gco.iarc.fr/today/data/factsheets/cancers/11-Liver-fact-sheet.pdf][doi: 10.3389/fonc.2020.00171] Please update the text accordingly.

Lines 56-60: I believe that the mention of the other treatments for HCC (such as TACE and RFA, see lines 108-118), should be moved here. Moreover, please be clearer regarding the current management of HCC, mentioning also the most widely adopted BCLC classification [doi: 10.1016/j.jhep.2021.11.018] The definitive therapies for HCC remain surgical resection and liver transplantation that can be performed only in patients at very early (0) and early (A) stages. However, given the similar survival benefit paired with the less invasiveness and lower costs compared to surgical resection, percutaneous ablative therapies such as radiofrequency ablation (RFA) and microwave ablation (MWA) are now considered the first treatment approach in both very early and early stages [doi: 10.1159/000521665][ doi: 10.1016/j.jhep.2013.04.009]. Despite inducing an effective local antitumor effect, the responses to ablation techniques are relatively weak and might not completely control the tumor, as testified by the high local recurrence rates [doi: 10.1002/bjs.7669]. Most patients with HCC (about 65-70%) are still diagnosed in the intermediate (B) or advanced (C) tumoral stages, thus resulting ineligible for radical therapies; therefore, patients with intermediate and/or advanced HCCs are considered for transarterial therapies or systemic therapies which, albeit effective, are deemed non-curative or “palliative” and still yield a lower 5-year survival rate. According to BCLC tumor staging and management, transarterial chemoembolization (TACE) is recommended as first-line therapy for unresectable intermediate-stage HCC (stage B) [doi: 10.3390/jpm11101041]. Recently, the role of other radiological locoregional therapies has expanded and other transarterial techniques, such as transarterial radioembolization (TARE) with yttrium-90, have been suggested as a safe and effective alternative treatment option for HCC patients with a liver-dominant disease who cannot tolerate systemic therapies [doi: 10.1016/S1470-2045(17)30683-6]. Please expand this topic and cite the aforementioned references.

Line 62: sorafenib is not the first-line drug for HCC. The combination of atezolizumab and bevacizumab is now regarded as the standard first-line treatment for patients with advanced HCC due to the significant and clinically meaningful improvements in terms of OS, PFS, objective response rate (ORR) and complete response rate (CRR) compared with sorafenib monotherapy [doi: 10.1016/j.jhep.2021.11.018] [doi: 10.1016/j.jhep.2021.11.030][ doi: 10.1056/NEJMoa1915745]. More recently, the combination of tremelimumab and durvalumab has been reported to be superior to sorafenib in patients with advanced or unresectable HCC, adding another first-line treatment option [doi: 10.1200/JCO.20.03555]. Please correct and make sure the same error is not present in other sections. 

Lines 83-84: “Thus, there is a strong need for cellular/molecular markers of treatment response.” Besides the need for markers of treatment response, due to the numerous restricted selection parameters, only 10–20% of HCC patients are eligible for first-line ICI therapy and this percentage is reduced to <10% in the second-line treatment. Therefore, considering that about 30-40% of them do not respond to these agents, only a small number of HCC patients could actually benefit from immunotherapy. Therefore, there is an urgent need for effective predictive serological and/or tissue biomarkers to identify patients likely to benefit from immunotherapy and thus dictate patient-specific therapy choices and reduce the economic burdens on healthcare systems; in addition, it would be possible to avoid ICI-associated adverse events in those patients identified as non-responders [doi: 10.3390/ijms24108598] [doi: 10.3390/cancers11111689]. Please expand this issue and cite the aforementioned references.

Lines 119-141 should be removed or, at least, drastically reduce. The main focus should be the gut microbiome and its interaction with ICI therapy in HCC. Moreover, please avoid expressions such as “comparable to the separation of Red sea waters of Moses”.

In chapter 3.2.1, please discuss only immunotherapy and leave its application to the HCC management and treatment in the subsequent chapter. For example, lines 157-159 and 168-170 should be moved to the following chapter. Accordingly, please rename 3.2.1 chapter.

Figure 1, please add also where ICIs act.

Chapter 3.2.2 should be moved before Chapter 3.2.1. Moreover, the expression “ “cancer-bug“ association” appears a bit odd.

Author Response

#Referee 2

Dear Authors

I would like to thank you for the opportunity to review this interesting paper focused on a very remarkable and challenging topic that is a lively argument also in daily clinical practice.

Immunotherapy has remarkably revolutionized the management of advanced HCC and prompted clinical trials, with therapeutic agents being used to selectively target immune cells rather than cancer cells. Despite the encouraging results, many unanswered questions still remain, including which immunotherapy and locoregional treatment can guarantee the best survival and clinical outcomes, the most effective timing and sequence to obtain the most effective therapeutic response and which biological and/or genetic biomarkers can be used to identify patients likely to benefit from this combined approach. In particular, considering the potential modulatory role of human microbiota on antitumoral immune response, the additional administration of specific probiotics as well as the more complex fecal microbiota transplantation in non-responder patients are currently being investi-gated, with rather promising preliminary results.

This paper is pleasurable to read, although it suffers from some limitations that Authors can easily adjust to slightly improve their review making it more eligible for this important Journal. Furthermore, the Authors can improve some sections of the paper, adding information and including other important references about this topic that, in my opinion, should be cited and discussed.

First of all, despite the title being correct, I suggest improving it. For example: “Perspective on the role of gut microbiome in the treatment of Hepatocellular carcinoma with Immune Checkpoint Inhibitors” or something similar.

We thank the reviewer for this suggestion. We have modified the title accordingly.

In the Introduction, line 46 “HCC is the fourth cancer’s cause of death worldwide”. This is not true. According to the most updated data, HCC is the sixth most commonly occurring cancer worldwide and, due to its constantly increasing incidence, has become the third leading cause of cancer-related death among general populations. Moreover, it represents the most common cause of death in patients with cirrhosis. [https://gco.iarc.fr/today/data/factsheets/cancers/11-Liver-fact-sheet.pdf][doi: 10.3389/fonc.2020.00171] Please update the text accordingly.

We thank the reviewer for this suggestion. We have modified the text accordingly.

Lines 56-60: I believe that the mention of the other treatments for HCC (such as TACE and RFA, see lines 108-118), should be moved here. Moreover, please be clearer regarding the current management of HCC, mentioning also the most widely adopted BCLC classification [doi: 10.1016/j.jhep.2021.11.018] The definitive therapies for HCC remain surgical resection and liver transplantation that can be performed only in patients at very early (0) and early (A) stages. However, given the similar survival benefit paired with the less invasiveness and lower costs compared to surgical resection, percutaneous ablative therapies such as radiofrequency ablation (RFA) and microwave ablation (MWA) are now considered the first treatment approach in both very early and early stages [doi: 10.1159/000521665][ doi: 10.1016/j.jhep.2013.04.009]. Despite inducing an effective local antitumor effect, the responses to ablation techniques are relatively weak and might not completely control the tumor, as testified by the high local recurrence rates [doi: 10.1002/bjs.7669]. Most patients with HCC (about 65-70%) are still diagnosed in the intermediate (B) or advanced (C) tumoral stages, thus resulting ineligible for radical therapies; therefore, patients with intermediate and/or advanced HCCs are considered for transarterial therapies or systemic therapies which, albeit effective, are deemed non-curative or “palliative” and still yield a lower 5-year survival rate. According to BCLC tumor staging and management, transarterial chemoembolization (TACE) is recommended as first-line therapy for unresectable intermediate-stage HCC (stage B) [doi: 10.3390/jpm11101041]. Recently, the role of other radiological locoregional therapies has expanded and other transarterial techniques, such as transarterial radioembolization (TARE) with yttrium-90, have been suggested as a safe and effective alternative treatment option for HCC patients with a liver-dominant disease who cannot tolerate systemic therapies [doi: 10.1016/S1470-2045(17)30683-6]. Please expand this topic and cite the aforementioned references.

We thank the referee for the detailed upgrade given. We have integrated these information within the suggested section. We have updated the suggested references also.

Line 62: sorafenib is not the first-line drug for HCC. The combination of atezolizumab and bevacizumab is now regarded as the standard first-line treatment for patients with advanced HCC due to the significant and clinically meaningful improvements in terms of OS, PFS, objective response rate (ORR) and complete response rate (CRR) compared with sorafenib monotherapy [doi: 10.1016/j.jhep.2021.11.018] [doi: 10.1016/j.jhep.2021.11.030][ doi: 10.1056/NEJMoa1915745]. More recently, the combination of tremelimumab and durvalumab has been reported to be superior to sorafenib in patients with advanced or unresectable HCC, adding another first-line treatment option [doi: 10.1200/JCO.20.03555]. Please correct and make sure the same error is not present in other sections.

We thank the reviewer for this useful observation. We have modified the text accordingly.

Lines 83-84: “Thus, there is a strong need for cellular/molecular markers of treatment response.” Besides the need for markers of treatment response, due to the numerous restricted selection parameters, only 10–20% of HCC patients are eligible for first-line ICI therapy and this percentage is reduced to <10% in the second-line treatment. Therefore, considering that about 30-40% of them do not respond to these agents, only a small number of HCC patients could actually benefit from immunotherapy. Therefore, there is an urgent need for effective predictive serological and/or tissue biomarkers to identify patients likely to benefit from immunotherapy and thus dictate patient-specific therapy choices and reduce the economic burdens on healthcare systems; in addition, it would be possible to avoid ICI-associated adverse events in those patients identified as non-responders [doi: 10.3390/ijms24108598] [doi: 10.3390/cancers11111689]. Please expand this issue and cite the aforementioned references.

We thank the reviewer for these interesting and precise observations and suggestions. We have updated the text accordingly, including the suggested references.

Lines 119-141 should be removed or, at least, drastically reduce. The main focus should be the gut microbiome and its interaction with ICI therapy in HCC. Moreover, please avoid expressions such as “comparable to the separation of Red sea waters of Moses”.

We have reduced this section and removed the cited expression, replaced by a more scientific sentence.

In chapter 3.2.1, please discuss only immunotherapy and leave its application to the HCC management and treatment in the subsequent chapter. For example, lines 157-159 and 168-170 should be moved to the following chapter. Accordingly, please rename 3.2.1 chapter.

Figure 1, please add also where ICIs act.

We thank the reviewer for this suggestion, we have changed the Figure accordingly.

Chapter 3.2.2 should be moved before Chapter 3.2.1. Moreover, the expression “ “cancer-bug“ association” appears a bit odd.

We thank the reviewer for this suggestion. Moreover, we have modified the title in more scientific way.

Round 2

Reviewer 2 Report

The Authors addressed raised points adequately.